

# COVID-19: molecular and serological detection methods

Ahmed E. Dhamad[1,2] and Muna A. Abdal Rhida[1,2]

[1] Cell and Molecular Biology, University of Arkansas at Fayetteville, Fayetteville, AR, USA
[2] Department of Biological Sciences, Wasit University, Kut, Wasit, Iraq

## ABSTRACT

Since COVID-19, caused by severe acute respiratory syndrome coronavirus 2 (SARS-CoV-2), was declared as a pandemic disease by the World Health Organization in early 2020, many countries, organizations and companies have tried to find the best way to diagnose the virus and contain its spreading. SARS-CoV-2 is a positive-sense single RNA (+ssRNA) coronavirus and mainly spreads through droplets, respiratory secretions, and direct contact. The early detection of the virus plays a central role in lowering COVID19 incidents and mortality rates. Thus, finding a simple, accurate, cheap and quick detection approach for SARS-CoV-2 at early stage of the viral infection is urgent and at high demand all around the world. The Food and Drug Administration and other health agencies have declared Emergency Use Authorization to develop diagnostic methods for COVID-19 and fulfill the demand. However, not all developed methods are appropriate and selecting a suitable method is challenging. Among all detection methods, rRT-PCR is the gold standard method. Unlike molecular methods, serological methods lack the ability of early detection with low accuracy. In this review, we summarized the current knowledge about COVID-19 detection methods aiming to highlight the advantages and disadvantages of molecular and serological methods.

## INTRODUCTION

In January 2020, World Health Organization (WHO) initially named a newly identified β-coronavirus that caused many pneumonia cases in December 2019 in Wuhan, China as the 2019-novel coronavirus (2019-nCoV) (*Zhu et al., 2020*; *Zhou et al., 2020*). Eventually, WHO and Coronavirus Study Group of International committee officially named the virus as SARS-CoV-2 and the disease as coronavirus disease 2019 (COVID-19) (*Guo et al., 2020*). SARS-CoV-2 is a member of the coronaviruses (CoV) family and it is an enveloped, non-segmented, positive-sense single RNA (+ssRNA) coronavirus (*Zhu et al., 2020*). In early 2020, the whole genome sequence of SARS-CoV-2 was revealed which was 29.9 kb (*Wu et al., 2020*) and 96.2% and 79.5% identical to a bat CoV RaTG13 and SARS-CoV genome sequences, respectively (*Zhou et al., 2020*; *Lu et al., 2020*). CoVs genome includes six to twelve open reading frames (ORFs) (*Song et al., 2019*). The first and largest ORF (ORF1a/b) occupies approximately two-thirds of the viral RNA

Corresponding author
Ahmed E. Dhamad,
adhamad@uark.edu

(*De Wilde et al., 2018*; *Cui, Li & Shi, 2019*) and the remaining one-third of the genome encodes the four main structural proteins which includes spike (S), envelope (E), membrane (M) and nucleocapsid (N) protein and other accessory proteins (*Cui, Li & Shi, 2019*; *Chen, Liu & Guo, 2020*; *Yang & Zhang, 2015*; *Zhang Lab, 2020*). The S protein plays a major role in SARS-CoV-2 infectious process and it is a promising target for vaccine and therapeutic development (*Jiang, Hillyer & Du, 2020*; *Du et al., 2009*).

COVID-19 virus is a highly contagious and spreads through droplets, respiratory secretions, and direct contact (*Cascella et al., 2020*; *Yang & Wang, 2020*). Recent studies reported that the virus was isolated from fecal swabs and blood samples of COVID-19 patients (*Zhang et al., 2020*; *Liu et al., 2020*) suggesting that the virus may have different routes to transmit between humans. The number of SARS-CoV-2 virus that causes ill to human is not clearly defined yet; however, a large hospitalized cohort ($n = 1,145$) was analyzed and the overall mean log10 viral load was 5·6 copies per mL (*Pujadas et al., 2020*). Elderly people and whom has chronic underlying diseases, such as but not limited to hypertension (*Kreutz et al., 2020*; *Schiffrin et al., 2020*), diabetes (*Hussain, Bhowmik & Do Vale Moreira, 2020*; *Fang, Karakiulakis & Roth, 2020*), and chronic obstructive pulmonary disease (*Lippi & Henry, 2020*), are the most vulnerable (*Muniyappa & Gubbi, 2020*; *Emami et al., 2020*). Current studies showed that the median age of COVID-19 patients was 47–59 years and females were the minority, less than 46% (*Wang et al., 2020*; *Guan et al., 2020*; *Li et al., 2020*). While children and youth have lower rates of COVID-19 infection compared to elder people (*Tian et al., 2020*; *Chan et al., 2020*; *Davies et al., 2020*). The incubation period of the virus is one to fourteen days with 3–7 days being the most (*Lauer et al., 2020*).

It has been reported that the clinical symptoms of confirmed COVID-19 patients were varied from mild flu-like symptoms to very severe respiratory symptoms and even respiratory and kidney failures and death (*Huang et al., 2020*; *Sohrabi et al., 2020*). According to WHO and other sources fever, dry cough and tiredness are the most common symptoms while sore throat, diarrhea, headache, conjunctivitis, rash on skin and discoloration of fingers or toes are less common symptoms of COVID-19 patients (*Wei et al., 2020*; *Kooraki et al., 2020*; *World Health Organization, 2020*). A recent study used an app-based symptom tracker showed that people who had COVID-19 loss of smell and taste and those with a positive test result (65.03%) intended to have anosmia higher than those with a negative test result (21.71%) (*Drew et al., 2020*; *Menni et al., 2020*). Although COVID-19 became a pandemic disease, the mortality rate is low (3.4%) compared to SARS and MERS patients, 9.6% and 35% respectively (*De Wit et al., 2016*).

The first incidents of COVID-19 were diagnosed in Wuhan, China, in December 2019. After a few months, WHO announced COVID-19 as a pandemic disease across the whole world. As of 2 July 2020, a total of 10,716,063 confirmed cases globally, 2,686,587 confirmed in USA and 8,029,476 outside of USA, with 516,726 globally deaths were reported by the Coronavirus Resource Center, Johns Hopkins University (*Coronavirus Resource Center, 2020*). Having a rapid and accurate diagnostic method at early stage of infection can help to contain the pandemic. Thus, many companies and laboratories were given authority under Emergency Use Authorization (EUA) restrictions to develop

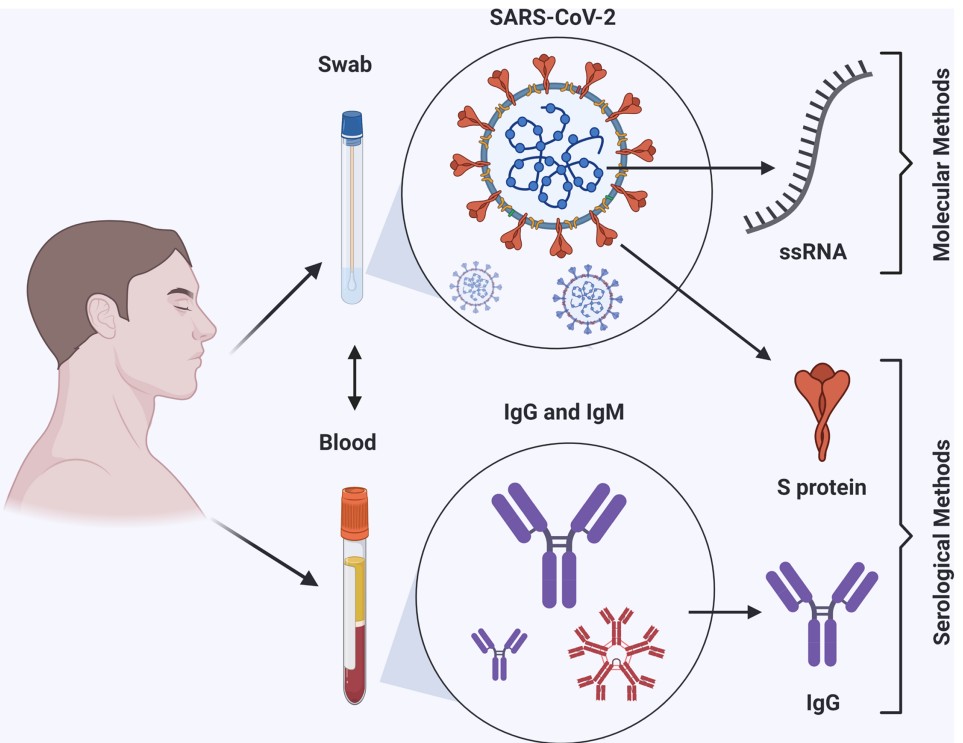

**Figure 1 Workflow summary of molecular and serological detection methods of COVID-19.**
The figure was created with BioRender.com.  

diagnostic methods. Consequently, hundreds of diagnostic kits based on different methods are available now, but selecting the proper method requires further investigation.
In this review, the standard and current molecular and serological detection methods for SARS-CoV-2 will be discussed and highlighted (Fig. 1). As of today, among all detection methods rRT-PCR is the gold standard method. Unlike molecular methods, serological methods lack the ability of early detection with low accuracy. This review intends to help health care providers and related branches to choose the appropriate method for battling the COVID-19 pandemic and rise the public knowledge about the methods that could be used to detect the virus.

## SURVEY METHODOLOGY

This literature review explored the peer-reviewed and preprint literatures with mainly focusing on COVID-19 disease and its molecular and serological detection methods. We searched the following databases and websites from March to July 2020: Google Scholar, PubMed, bioRxiv, medRxiv, I-TASSER, CDC, WHO, Coronavirus Resource Center (Johns Hopkins University), Chinese Center for Disease Control and Prevention (CCDC) and National Institute of Infectious Diseases (NIID). And the top keywords that searched were: COVID-19, SARS-CoV-2, coronavirus, genomic RNA, protein structure, ACE2, transmission, symptoms, molecular detection methods, serological detection methods, rRT-PCR, ID NOW COVID-19, isothermal amplification, CRISPR,

SARS-CoV-2 DETECTR, LAMP, recombinase polymerase amplification (RPA), Lateral flow assay (LFA) and Enzyme-linked immunosorbent assay (ELISA).

## Diagnostic methods

Under the pressure of the pandemic, COVID-19 test demand is sharply increased which pushes a lot of biotech companies/ inventors to produce different kits based on variant approaches to detect SARS-CoV-2. The molecular and serological methods are the main methods to detect the virus.

## Molecular methods

Based on how viral RNA be processed and detected, there are three major molecular methods which are: real-time reverse transcription polymerase chain reaction (rRT-PCR), isothermal amplification, and clustered regularly interspaced short palindromic repeats (CRISPR) based methods. All these methods follow the same protocol that have been recommended by the Centers for Disease Control and Prevention (CDC) for collecting specimens from COVID-19 patients (*Centers of Disease Control and Prevention (CDC), 2020*).

## rRT-PCR method

It is the gold standard and reliable molecular method to diagnose SARS-CoV-2 with high sensitivity (positive agreement) and specificity (negative agreement) (*Corman et al., 2020*). This method has been developed by several laboratories to detect COVID-19 virus (*Amrane et al., 2020*; *Capobianchi et al., 2020*; *Chu et al., 2020*). In this method (Fig. 2B), cDNA is generated from the extracted RNA of COVID-19 virus with specific primers for the following genes 2019nCoV-N1 (N1), 2019nCoV-N2 (N2) and RNAse P (RP; internal control) as recommended by U.S. CDC (Table 1) and other health agencies (*Corman et al., 2020*; *Niu et al., 2020*; *Nao et al., 2020*) (Table S1). The upper respiratory system's swabs are the main specimens that are used to detect COVID-19 virus; however, serum, ocular secretions and stool can be used as well (*Xia et al., 2020*; *Carter et al., 2020*; *The COVID-19 Investigation Team, 2020*). If both genes (N1 and N2) were positive, it is considered as a positive sample as shown in Table 2. The positive result confirms the presence of viral RNA in the specimen, but not necessarily the virus viability (*Sethuraman, Jeremiah & Ryo, 2020*). Besides the internal control (RP), there are three controls that must be run to make sure the result is legitimate (Table S2). These controls are 2019-nCoV Positive Control (nCoVPC), No Template Control (NTC) and Human Specimen Control (HSC) (*Centers of Disease Control and Prevention (CDC), 2020*). Even though rRT-RPC is the gold standard method and the most widely used for diagnosing COVID-19 virus in clinic and research laboratories, it has some limitations (*Kubina & Dziedzic, 2020*). Beside highly costed, professional skills needed, it is time-consuming (requires 2–5 days from collecting a sample till getting the result) and must be done in a laboratory.

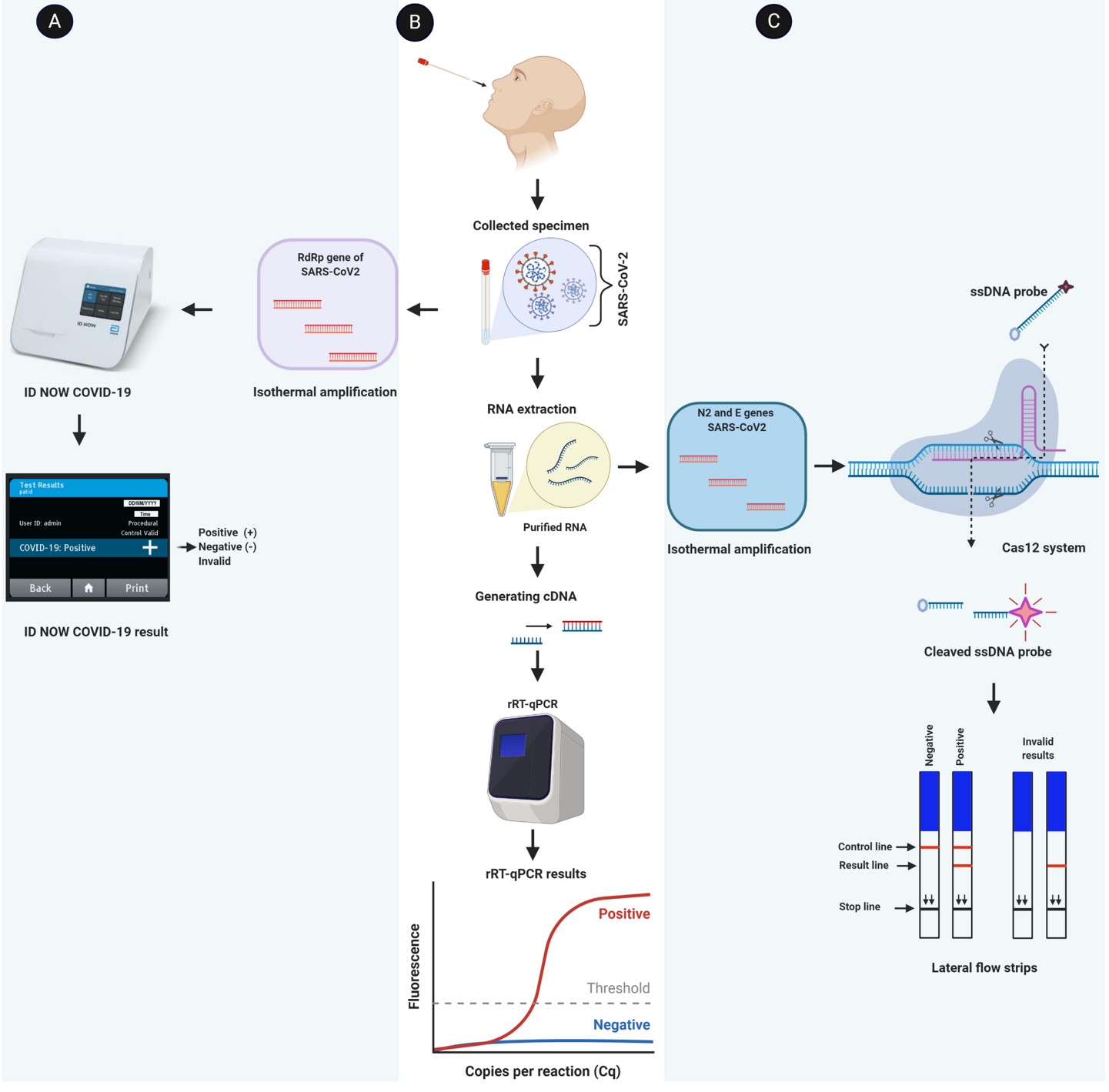

**Figure 2 Schematic flowchart of molecular detection methods for COVID-19 virus.** (A) A viral RNA is directly amplified and detected from the sample by isothermal amplification method (e.g., ID NOW COVID-19). (B) The standard rRT-PCR method is illustrated where a viral RNA is extracted and converted into cDNA. Specific areas of cDNA (target genes) are amplified and detected by rRT-PCR. (C) Demonstration of CRISPR-Cas12 based method. After amplifying specific areas of extracted viral RNA, the Cas12 enzyme recognizes specific sequences and then cleaves the ssDNA probe. The cleavage probes are visualized as a red line in the lateral flow strip. The figure was created with BioRender.com.

**Table 1 Primers and probes that have been recommended by the U.S.CDC to detect SARS-CoV-2 by rRT-PCR.**

| Gene target | Description | Oligonucleotide Sequence (5′>3′) | Label |
|---|---|---|---|
| 2019-nCoV_N1 | 2019-nCoV_N1 Forward Primer | 5′-GAC CCC AAA ATC AGC GAA AT-3′ | None |
| | 2019-nCoV_N1 Reverse Primer | 5′-TCT GGT TAC TGC CAG TTG AAT CTG-3′ | None |
| | 2019-nCoV_N1 Probe | 5′-FAM-ACC CCG CAT TAC GTT TGG TGG ACC-BHQ1-3′ | FAM, BHQ-1 |
| 2019-nCoV_N2 | 2019-nCoV_N2 Forward Primer | 5′-TTA CAA ACA TTG GCC GCA AA-3′ | None |
| | 2019-nCoV_N2 Reverse Primer | 5′-GCG CGA CAT TCC GAA GAA-3′ | None |
| | 2019-nCoV_N2 Probe | 5′-FAM-ACA ATT TGC CCC CAG CGC TTC AG-BHQ1-3′ | FAM, BHQ-1 |
| RNAse P | RNAse P Forward Primer | 5′-AGA TTT GGA CCT GCG AGC G-3′ | None |
| | RNAse P Reverse Primer | 5′-GAG CGG CTG TCT CCA CAA GT-3′ | None |
| | RNAse P Probe | 5′-FAM—TTC TGA CCT GAA GGC TCT GCG CG—BHQ-1-3′ | FAM, BHQ-1 |

**Table 2 Expected results and their interpretations of rRT-PCR method for COVID-19 specimens.**

| 2019nCoV-N1 | 2019nCoV-N2 | RNaseP | Result (SARS-CoV-2) |
|---|---|---|---|
| + | + | +/− | Positive |
| If only 1 of the 2 targets is positive | | +/− | In conclusive |
| − | − | + | Negative |
| − | − | − | Invalid |

## Isothermal amplification-based method

It is another molecular approach where a nucleic acid is rapidly and specifically amplified by a polymerase with high strand displacement activity (e.g., optimized Bst polymerase) and different sets of primers at constant temperature (60–65 °C) without the need of thermal cycler (*Notomi et al., 2000*). ID NOW COVID-19 (Abbott) is a recent example of using isothermal amplification technique to detect COVID-19 virus in clinics. It is molecular point-of-care platform in the United States of America and used under an EUA only to diagnose SARS-CoV-2. In this test a certain region of RdRp gene of SARS-CoV-2 is amplified by specific primers and results are displayed in a short time compared to rRT-PCR (Fig. 2A). It can show a positive result as little as 5 min and a negative result in 13 min. It has a performance of ≥94% sensitivity and ≥98% specificity compared to lab-based PCR reference tests as it is advertised by the manufacturers (*Abbott, 2020*). However, *Harrington et al. (2020)* results that published in a peer-reviewed journal showed that the overall sensitivity and specificity were 74.73% and 99.41%, respectively.

## CRISPR-Cas12 based method

In this method (e.g., SARS-CoV-2 DETECTR), the RNA virus is extracted from a specimen and designated regions of N2, E, RP genes are amplified at 62 °C for 20 min by specific primes through Reverse Transcription Loop-mediated Isothermal Amplification (RT-LAMP) approach (*Wang et al., 2020*; *Lamb et al., 2020*; *Hong et al., 2004*). Then, designed Cas12 gRNAs direct Cas12 protein to specific areas of the above amplified

**Table 3 Expected results and their interpretations of SARS-CoV-2 DETECTR method for COVID-19 specimens.**

| N gene | E gene | RNaseP | Result (SARS-CoV-2) |
|--------|--------|--------|---------------------|
| + | + | +/− | Positive |
| + | − | +/− | Indeterminate |
| − | + | +/− | Indeterminate |
| − | − | + | Negative |
| − | − | − | Invalid |

genes where a reporter molecule (a single stranded DNA (ssDNA) probe) is cleaved. This reaction occurs at 37 °C for 10 min and the result is visualized by a fluorescent reader or a lateral flow strip (Fig. 2C). Both genes N2 and E must be positive to consider the sample is positive (Table 3). *Broughton et al. (2020)* showed that SARS-CoV-2 DETECTR was reliable to detect coronavirus in respiratory swab samples with 90% sensitivity and 100% specificity. Unlike rRT-PCR, this method is fast (<50 min), cheap, and point-of-care test (POCT). It requires less equipment and the result can be visualized by naked eyes. However, it requires troubleshooting and specific design of all enzymes, primers, and reporters that are used in this method.

In addition to the above molecular methods, Recombinase polymerase amplification (RPA) (*Amer et al., 2013*) has been developed and/or integrated with other methods to detect COVID-19 virus. This method does not require thermal cycler and can be used as POCT with low cost and high sensitivity and specificity. The drawback is that it requires several specific designed primers which could be difficult to obtain and the result of this method could be interfered by virus quantification and debris (*Yu et al., 2020*).

## Serological methods

Unlike molecular methods, serological methods (also called antibody tests) can be applied to detect past and current SARS-CoV-2 infection and monitor the progress of the disease periods and immune response. They can detect the presence of antibodies (e.g., IgG, IgM and IgA) in a COVID-19 patient's serum and plasma. Other biological fluids such as but not limited to saliva and sputum could be used as well. Antibodies are produced as a defense mechanism by the immune system against SARS-CoV-2. First, IgM is produced after a few days of infection and last for approximately two weeks which followed by IgG production that is last longer (*To et al., 2020*; *Xiang et al., 2020*). Thus, detecting IgM in a patient's sample indicates early-stage infection while detecting IgG indicates a current or prior infection (*Sethuraman, Jeremiah & Ryo, 2020*). In addition to lacking an early detection, accuracy is the main challenge of these approaches where crossover could occur with other antibodies that produced as a result of infection of other coronavirus family members such as SARS-CoV (*U.S. Food & Drug Administration, 2020*; *Maxim, Niebo & Utell, 2014*).

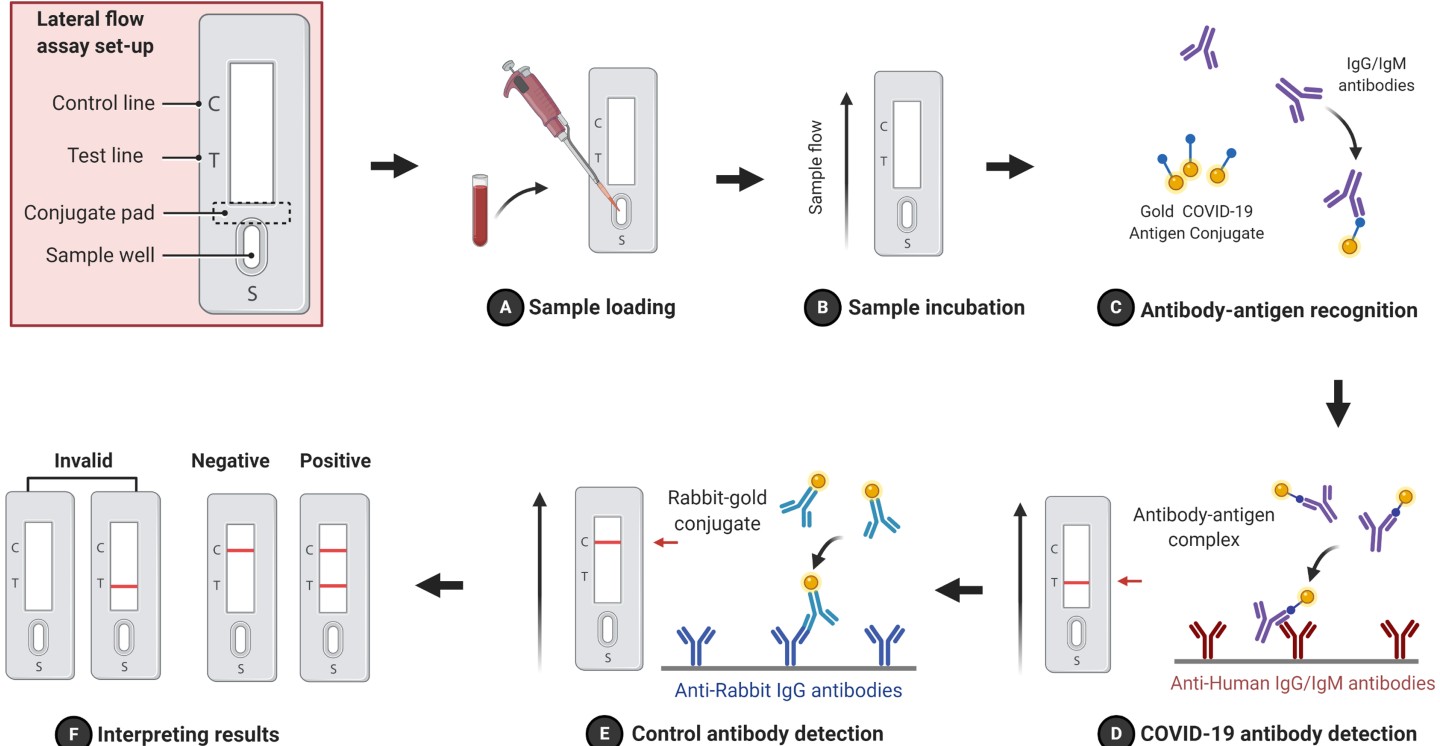

**Figure 3 Schematic flowchart of FLA.** A sample is loaded in the sample well (A) and incubated to allow the capillary action to move sample antibodies (IgG/IgM) forward (B). Gold COVID-19 antigen conjugates from the conjugate pad recognize and interact with sample antibodies forming complexes (C) that are immobilized by anti-human IgG/IgM antibodies and display the test red line (D). Control antibodies (rabbit- gold conjugates) are immobilized by anti-rabbit IgG antibodies and show the control red line (E). (F) FLA results possibilities are illustrated. The figure was created with BioRender.com.                                

## Lateral flow assay

It is one of the most popular serological method that has been applied in clinics to detect antigens (*Boisen et al., 2015*), antibodies (*Nielsen et al., 2008*), and amplified nucleic acids (*Koczula & Gallotta, 2016*; *Rohrman et al., 2012*) in variant biological samples such as blood (serum or plasma) (*Schramm et al., 2015*; *Magambo et al., 2014*), urine (*Moreno et al., 2017*) and saliva (*Carrio et al., 2015*). LFA is a paper-like membrane strip that is coated with two lines. The first line, the test line, contains anti-human IgG/IgM antibodies, while the second line, the control line, contains anti-rabbit IgG antibodies. After adding a patients specimen (e.g., blood) into the sample well, IgG/IgM antibodies are moved by capillary action toward the lines crossing through the conjugated pad where a specific conjugated antigen (e.g., gold COVID-19 antigen conjugate) and rabbit-gold conjugated antibodies are impeded (*Parolo, De la Escosura-Muniz & Merkoci, 2013*). IgG/IgM antibodies are interacted and made a complex with gold COVID-19 antigen conjugate. The complex binds anti-human IgG/IgM antibodies and immobilizes at the test line, while the rabbit-gold conjugate antibodies bind anti-rabbit IgG antibody and immobilized at the control line. The result will be visible as a red line due to the accumulation of gold particles. If both test and control lines appear red, the result is positive and negative when only the control line appears red. If both lines disappear or only the test line appears,

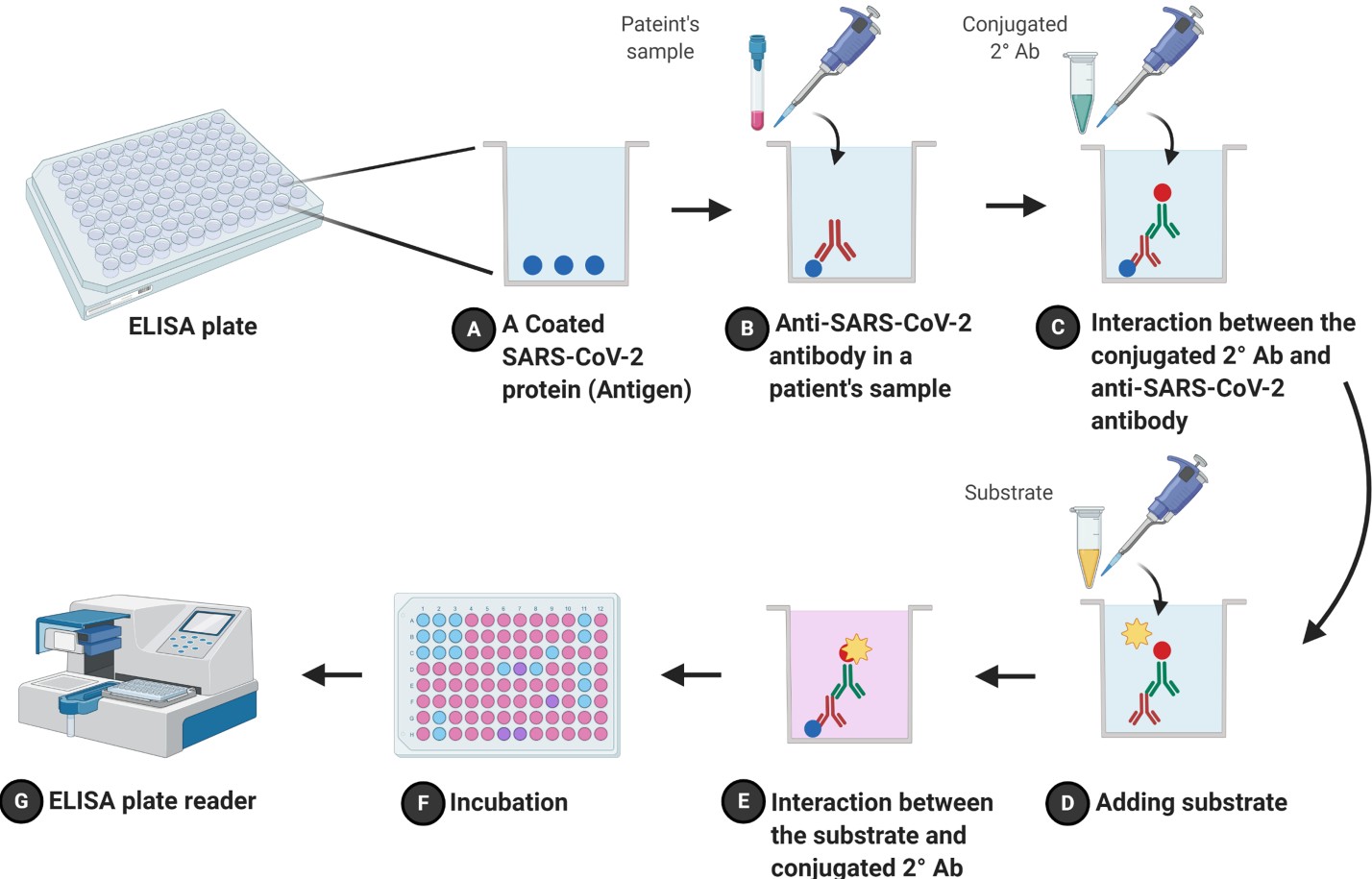

**Figure 4 Schematic flowchart of indirect ELISA.** A coated SARS-CoV-2 protein (antigen) onto wells of ELISA plate (A) interacts with the first antibody (anti-SARS-CoV-2 antibody) that is in a patient's sample (B). (C) After adding a secondary antibody (a conjugated antibody), it recognizes and interacts with the first antibodies. The reaction is developed by adding a substrate (D) which is cleaved by the conjugated enzyme and changes the reaction color after incubation (E) and (F), respectively. (G) Results are read by ELISA plate reader. The figure was created with BioRender. com.                               

the result is invalid (Fig. 3). The advantages of FLA are rapid (10–30 min), cheap, no need for professional skills and portable (POCT). It can be done by 1–2 blood drops and the result is visualized by naked eyes without an expensive equipment. The drawback of FLA is a qualitative method, tells the presence or absence of antibodies against the virus without telling how much they were in a patient's sample, and it less accurate compared to rRT-PCR. It was showed that FLA has clinical 57% sensitivity and 100% specificity for IgM and 81% sensitivity and 100% specificity for IgG (*Xiang et al., 2020*).

## Enzyme-linked immunosorbent assay

It is another serological method and called enzyme immunoassay (EIA). ELISA is a plate-based method that has been used for detecting and quantifying soluble substances such as proteins and antibodies in clinic and research laboratories. It includes direct and indirect formats (*Zhang et al., 2014*). The indirect ELISA, the most popular and more sensitive than the direct ELISA, an antigen (e.g., a recombinant protein (N protein) of

**Table 4 Comparison between molecular and serological methods for detecting COVID-19 virus.**

|  | Molecular methods | | | Serological methods | |
|---|---|---|---|---|---|
| Technique based | rRT-PCR | Isothermal amplification | CRISPR-Cas12 | LFA | ELISA |
| Sample | RNA | RNA | RNA | Ag or Ab | Ag or Ab |
| Accuracy | High | High/Moderate | High | Low | Moderate |
| Time* | Hours | Minutes | Minutes | Minutes | Hours |
| Professional skills need | Yes | Yes | YES/No | No | Yes |
| POCT | No | Yes/No | Yes/No | Yes | No |
| Availability | Limited | Limited | Limited | Available | Available |
| Cost | Very high | High | Average | Low | Average |
| High throughput | Yes | No | No | No | Yes |

**Note:**
 * Time that is required for running the test without preparation time.

SARS-CoV-2 virus) is coated onto the inner surface of 96-well or 384-well polystyrene plates (*Gao et al., 2015*). A diluted patient's plasma which may have anti-SARS-CoV-2 IgG/IgM is added to the wells. The plate is incubated for one hour to allow the antibodies to interact with coated antigens. After washing the plate to eliminate unspecific interactions, a conjugated antibody with a reported enzyme such as horseradish peroxidase (HRP) or alkaline phosphatase (AP) is added to form sandwich complexes (*Li et al., 2013*; *Zhang et al., 2015*). These complexes are detected and quantified by adding a substrate (e.g., 3,3′,5,5′-tetramethylbenzidine) that is utilized by the report enzyme and leads to change in the reaction color (*Madersbacher & Berger, 1991*; *Lee, Harrison & Lewis, 1990*). The color is detected and measured by a plate reader (Fig. 4). ELISA is relatively fast (2–5 h) and cheap compared to rRT-PCR, and it is similar to FLA regard to accuracy. It has been reported that ELISA results were 50% (IgG) and 81% (IgM) for patients on day zero and became 81% (IgG) and 100%(IgM) on day five of SARS-CoV-2 infection (*Zhang et al., 2020*). Another study accomplished by *Xiang et al. (2020)* showed that using ELISA to detect IgM and IgG on day four of symptom onsite revealed a sensitivity of 77.3% and specificity of 100% for IgM while those were 83.3% and 95% respectively for IgG.

Worth mention that there are other serological methods that are less common than FLA and ELISA (*La Marca et al., 2020*). A colloidal gold immunochromatography assay (GICA), and Chemiluminescent immunoassay (CLIA) were developed to diagnose COVID-19; however, they have low sensitivity at the beginning of the infection (*Infantino et al., 2020*; *Zhang et al., 2020*). *Pan et al. (2020)* reported that the sensitivity of GICA were 11.1% on the first week and 92.9% on the second weeks after the onset of symptoms. Neutralization assays, on the other hands, are standard methods for determining antibody efficacy (e.g., serum virus neutralization (SVN) assay). They are used to check whether a patient has active antibodies that can neutralize the SARS-CoV-2 infection (*Gauger & Vincent, 2020*; *Muruato et al., 2020*). These assays play a key role in determining if an individual is eligible to donate his/her convalescent plasma as a treatment for seriously ill people although such treatment has not been fully validated (*Shen et al., 2020*).

Both molecular and serological methods are not perfect in terms of detecting COVID-19 virus and each method has its own limitations (*Bisoffi et al., 2020*). Though molecular methods are more reliable than serological methods, both methods could give false results due to various reasons. For instance, incorrect sampling, inadequate viral material in the specimen, improper RNA extraction, cross-reactions with other viral species, contamination and technical issues could lead to positive and negative false results. To overcome such issues and increase the certainty of given results, these methods can be followed by secondary diagnostic methods such as a chest CT scan and x-ray imaging (*Yang et al., 2020*; *Wang et al., 2020*; *Wong et al., 2020*).

## CONCLUSION

Scientists have made significant progress in the characterization of the COVID-19 virus and how to limit its spreading. Also, they are working hard on diagnostic methods and finding therapies and vaccines against the virus. Currently, neither an approved vaccine nor a specific antiviral treatment is available for COVID-19 disease. Thus, detecting SARS-CoV-2 at the early infectious stage by a rapid and accurate diagnostic method could save thousands of lives. In this review we have discussed and summarized the current knowledge about molecular and serological methods that have been used to detect SARS-CoV-2. Though the molecular methods are more expensive, slower, and less available than serological methods, they are more accurate and rRT-PCR is the gold standard method among them (Table 4). Further research and collaboration between scientists and companies are needed to overcome some limitations of current methods and might find a new and better avenue to detect the virus. For instance, standardized the methods, produce new and high-quality kits and make them available at low cost will make the current methods more reliable.

### Funding
The authors received no funding for this work.

### Competing Interests
The authors declare that they have no competing interests.

### Author Contributions
- Ahmed E. Dhamad conceived and designed the experiments, performed the experiments, analyzed the data, prepared figures and/or tables, authored or reviewed drafts of the paper, and approved the final draft.
- Muna A. Abdal Rhida performed the experiments, analyzed the data, prepared figures and/or tables, authored or reviewed drafts of the paper, and approved the final draft.

### Data Availability
    This is a review article; there is no raw data.

## Supplemental Information

Supplemental information for this article can be found online at http://dx.doi.org/10.7717/peerj.10180#supplemental-information.

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
