# Peer review of "COVID-19: molecular and serological detection methods"

_PeerJ, doi:10.7717/peerj.10180_

## Round 0.1 · original submission · Major Revisions

I think that you should pay attention to all reviewers' comments. Especially, these two ones: "Language is at times imprecise" and "...parts on biology of SARS-COV-2 coronavirus, which are commonplace, should be removed... this review is too general, and, should be shortened".

Reviewer 1 ·

Basic reporting

Is the review of broad and cross-disciplinary interest and within the scope of the journal?
The review proposes a description of the SARS-CoV-2 virus genome, a summary of transmission mechanisms, and an overview of direct and indirect methods of viral infection detection.
The topic is of great interest, broad and interdisciplinary, and within the scope of the journal.

Has the field been reviewed recently? If so, is there a good reason for this review (different point of view, accessible to a different audience, etc.)?
Reviews are of great interest in order to help readers to make sense of the large information content present in the scientific literature. Since the number of publications related to COVID-19 is extremely high and growing fast, reviews published within few months have a large amount of novel information to discuss. That said, an example of a review that discusses diagnostics approaches for COVID-19 is https://doi.org/10.3389/fcell.2020.00468. The value of this published review is that it clearly lists advantages and disadvantages of every approach discussed and clearly differentiates test used in the clinics from experimental assays that are under development.
Does the Introduction adequately introduce the subject and make it clear who the audience is/what the motivation is?
The audience is clearly identified. Both the abstract and the introduction do not set a clear expectation for the following sections: “SARS-CoV-2 structure” and “Transmission and symptoms of COVID-19”. If the sections are to be retained, they should be announced in the introduction and in the abstract.

Experimental design

Is the Survey Methodology consistent with a comprehensive, unbiased coverage of the subject? If not, what is missing?
Survey Methodology clearly describes the databases used for literature research and the keywords used in the search. It would be useful to state the date the searches were conducted. A nice addition to the diagnostic methods might have been a discussion on neutralization methods.
Are sources adequately cited? Quoted or paraphrased as appropriate?
Sources are adequately cited and paraphrased. The bibliography includes 104 references.
Is the review organized logically into coherent paragraphs/subsections?
The review is logically organized and easy to follow. More emphasis of advantages and disadvantages of every testing approach and a clear delineation of what tests are currently used in the clinics and what assays are experimental would have been very helpful.
The paragraphs on the structure of the virus and on transmission and symptoms need to be announced in the Introduction and Abstract. On transmission, current estimation of the number of viruses needed to infect a human host might be relevant and interesting to the readership. A stratification of symptoms according to the WHO might be helpful.

Validity of the findings

Is there a well developed and supported argument that meets the goals set out in the Introduction?
The introduction states that the goal of diagnostic efforts is to achieve a rapid and accurate diagnostic method. Nowhere in the review it is stated that serological methods are not fulfilling the scope of early detection because seroconversion happens weeks after infection. Antibody measurement indicates exposure to the virus.
It is also very important to clarify the concept of “serological method” that in the present review appears to be detection of proteins in serum. Antibody measurement indicates exposure, viral antigen testing indicates presence of the virus and likely active infection. This very important concept needs to be clarified throughout the manuscript and in the abstract. If a discussion of serology diagnostic methods is retained in the manuscript, the goal in the abstract and introduction of rapid testing needs to be rephrased and expanded.
Does the Conclusion identify unresolved questions / gaps / future directions?
The conclusion is that further research is needed to overcome present limitations of diagnostic approaches but it is not very specific nor lays out future directions. Interesting points might be: including need for standardization, need of coordination with regulatory agencies, accessibility, reagent quality are not discussed. Additionally, the use of testing in a scenario where a vaccine is available would be interesting.

Additional comments

General comments.
The review covers recent information about the SARS-CoV-2 structure, COVID-19 symptoms and transmission, and diagnostic tests including 1) PCR and other nucleic acid amplification technologies, 2) serology, and 3) direct antigen test. The intended audience is clearly defined. The authors compiled a good amount of information on the testing approaches currently available.

Major issues with the manuscript (issues are listed in order of importance):
- Serology tests do not include direct antigen tests in serum. The concept of serological tests should be clarified.
- The need for COVID-19 testing should be clarified in the abstract and introduction. If the goal is to achieve early diagnosis, antibody testing is not the preferred test.
- Tests should be critically discussed and advantages and disadvantages including costs, turnaround time, sensitivity, specificity should be presented. The fact that PCR is the gold standard for virus detection should be stated.
- Introduction should mention that a description of the virus and its symptoms is present.
- A distinction between clinically used tests and research assays should be clearly defined.
- Language is at times imprecise.

Reviewer 2 ·

Basic reporting

this paper represents good academic work on summarizing existing techniques to detect SARS-CoV-2 in real life samples collected form patients or prospective patients. This field has been extensively reviewed recently, for example in examples below. Many of the published review go into much further details on technologies reviewed, and provide a very detailed landscape of futuristic trends, which current review lacks. Also, this review is too general, and contain parts on biology of SARS-COV-2 coronavirus, which are commonplace, and should be removed

1. PMID: 32729549
2. PMID: 32729494
3. PMID: 32641875
4. PMID: 32609256
5. PMID: 32607246

Experimental design

Also, this review is too general, and contain parts on biology of SARS-COV-2 coronavirus, which are commonplace, and should be removed. Sources are cited adequately, but some parts are present in incorrect sections, i.e. LAMP PCR is reviewed in two different sections, depending on subtype of the method

Validity of the findings

I think this review is too general, and, because of that, short of expectations.

Additional comments

nice effort, but not specific (focused) enough for PeerJ

---

## Round 0.2 · Minor Revisions

I must apologize for so long period of time that our reviewers are taking to provide me with their decisions. I am sorry but because of so long waiting time I must ask you to include in your literature review several additional publications. I hope that you will be able to make these changes quickly.

Molecular and Serological Tests for COVID-19 a Comparative Review of SARS-CoV-2 Coronavirus Laboratory and Point-of-Care Diagnostics
R Kubina, A Dziedzic - Diagnostics, 2020

Antonio La Marca, et al., Testing for SARS-CoV-2 (COVID-19): a systematic review and clinical guide to molecular and serological in-vitro diagnostic assays. Reproductive BioMedicine 41, 483-499, 2020

Bisoffi et al.Sensitivity, Specificity and Predictive Values of Molecular and Serological Tests for COVID-19: A Longitudinal Study in Emergency Room - Diagnostics 2020

---

## Round 0.3 · accepted · Accept

Congratulations! Thank you for your efforts.